# Review of Adoptive Cellular Therapies for the Treatment of Sarcoma

**DOI:** 10.3390/cancers17081302

**Published:** 2025-04-12

**Authors:** James J. Fradin, John A. Charlson

**Affiliations:** 1Division of Medicine, Medical College of Wisconsin, Milwaukee, WI 53226, USA; 2Division of Hematology & Oncology, Medical College of Wisconsin, Milwaukee, WI 53226, USA; jcharlso@mcw.edu

**Keywords:** sarcoma, adoptive cellular therapy, chimeric antigen receptor, tumor infiltrating lymphocyte, engineered T-cell receptor

## Abstract

Sarcomas are rare cancers with limited treatment options, especially when they metastasize. Adoptive cellular therapies, which use specially engineered immune cells to fight cancer, are a new and exciting field for the treatment of these otherwise resistant diseases. Early studies have shown promise, with some products already receiving FDA treatment approvals. If successful, these novel therapies could provide effective and long-lasting treatment for sarcoma patients who currently have few alternatives. Further clinical trials and combination with other therapies may significantly improve outcomes and change the way sarcomas are treated.

## 1. Introduction

Sarcomas represent less than 1% of all new cancer cases diagnosed yearly, with an incidence of 5 cases in 100,000 [1]. Sarcomas and soft-tissue tumors fall into a number of classifications: adipocytic tumors (including myxoid liposarcoma), fibroblastic/myofibroblastic tumors, fibrohistiocytic tumors, vascular tumors, perivascular tumors, smooth muscle tumors, skeletal muscle tumors, gastrointestinal stromal tumors, chondro-osseous tumors, peripheral nerve sheath tumors, and tumors of uncertain differentiation (including synovial sarcoma). More broadly, they are typically classified as bone sarcomas and soft-tissue sarcomas. Diagnosis is based on histology; immunohistochemistry; and, increasingly, genetic features, as a significant number of sarcoma subtypes are associated with specific chromosomal translocations. Synovial sarcoma, for example, is associated with translocation t(X;18).

Bone and soft-tissue sarcomas may be treated and even cured when diagnosed at a localized stage, and some sarcomas, such as Ewing’s sarcoma and osteosarcoma, may be put into long-term remission by multi-modality treatment, even if they present with metastatic disease. When sarcomas recur in a disseminated manner, they are generally not curable, and there are few therapies that effectively control the disease. The median survival from diagnosis of metastatic sarcoma is 18–24 months, and treatment is most often a repeated cycles of chemotherapy [2,3].

Immunotherapy, primarily with immune checkpoint inhibitors (ICIs), has resulted in significant improvement in the duration and quality of survival for patients with a variety of metastatic solid tumor types including melanoma, lung cancer, bladder cancer, and others. A favorable side effect profile and the potential for long-lasting tumor response are appealing features of immunotherapy. Sarcoma treatment with ICIs has not been as successful. Several subtypes of sarcoma, including undifferentiated pleomorphic sarcoma, angiosarcoma, and alveolar soft parts sarcoma, have demonstrated responsiveness to ICIs in clinical trials, but other sarcoma subtypes are mostly resistant to ICIs [4,5,6]. Sarcomas have a relatively lacking immune cell infiltrate and low tumor mutation burden, thus rendering them immunologically “cold” [7].

ACT is a type of immunotherapy in which immune cells (most commonly T-cells) are isolated from a patient’s blood or tumor, sometimes modified to enhance their cancer killing ability, and expanded in a lab before being given back to the patient. These technologies have shown promise in generating anti-tumor immune responses in non-immunogenic sarcomas.

## 2. Tumor Infiltrating Lymphocytes (TILs)

TILs are created by isolating lymphocytes from a patient’s tumor, then expanding them in vitro. The cells are then reinfused back into the patient (Figure 1) [8]. Typically, TILs will be infused following lymphodepleting chemotherapy, and infusion can be followed by the administration of IL-2, which has been shown to result in improved tumor regression in some malignancies [9].

TIL-based therapy has shown some success in clinical trials for both melanoma and multiple myeloma [10,11]. There are barriers to the use of TIL-based therapy in sarcoma, including poor T-cell infiltration into tumors, and an immunosuppressive tumor microenvironment (TME) [12]. T-cell numbers may be increased using in vitro rapid expansion protocols after T-cells have been harvested from tumors in an attempt to counteract this immunologically cold environment [13]. Additional strategies may be employed to increase T-cell infiltration. The ATLAS-IT-04 trial was a pilot study of LTX-315 along with TILs for the treatment of advanced or metastatic soft-tissue sarcoma (STS) [14]. LTX-315 is an intratumoral oncolytic peptide that induces necrosis and T-cell infiltration [15]. Six patients were included in the trial: two with desmoplastic round cell tumors, two with solitary fibrous tumors, one with leiomyosarcoma, and one with sclerosing epithelial fibrosarcoma. At 6 weeks post-infusion, 75% of the patients had stable disease (SD), and the mean progression free survival (PFS) was 22 weeks (range, 10–30). In terms of adverse events, all patients experienced grade 4 neutropenia and thrombocytopenia related to lymphodepleting chemotherapy, and many had grade 3 fevers related to TIL infusion. Another clinical trial investigating the use of TILs in adolescents and young adults with metastatic, high-grade STS reported that none of the nine recruited patients had objective anti-tumor response by RECIST criteria at 12 weeks post-infusion, but five of five assessable patients did have detectable circulating TILs at 6 weeks post-infusion (NCT04052334).

Although the results of the ATLAS-IT-04 trial are encouraging for the use of TILs in the future treatment of sarcomas, there is more work to be done. There is a striking lack of data for effective TIL-based therapies for sarcomas. The application of additional strategies like co-infusing TILs with other therapies (such as vaccines or other systemic chemotherapy and/or immunotherapies) or even genetically engineered TILs could improve these metrics. These challenges and questions require additional clinical trials (Table 1).

## 3. Chimeric Antigen Receptor (CAR) T-Cells

CARs are engineered receptors targeting specific cell surface epitopes that are genetically grafted to immune cells (most commonly T-cells) (Figure 2). They preferentially target tumor cells expressing the target antigen, increasing the anti-tumor activity of associated T lymphocytes [16]. One advantage of CAR T-cell therapy is the ability to target cell-surface antigens and markers on tumor cells without HLA restrictions, which limit other cellular therapies. However, this means CAR T-cells cannot target intracellular antigens expressed by peptide-MHC complexes. As with TILs, CAR T-cells are often infused following lymphodepleting chemotherapy.

While CAR T-cells have proven effective in the treatment of hematologic malignancies, the same efficacy has not been reported for solid tumors [17]. Although the exact cause of this discrepancy is unknown, proposed mechanisms include the effect of the immune suppressive TME, poor CAR T-cell movement to tumor sites, and antigen downregulation [17].

Human epidermal growth factor receptor 2 (HER2) is an antigen that is overexpressed in some sarcomas and may be associated with worse prognosis in osteosarcoma [18]. Unfortunately, clinical benefit has not been noted in several small clinical trials treating osteosarcoma with the HER2-directed monoclonal antibodies trastuzumab and trastuzumab deruxtecan [19,20]. This lack of efficacy may be due, at least in part, to low density of HER2 expression. In contrast, a study in osteosarcoma mouse models showed tumor regression after treatment with HER2-specific CAR-T cells [21].

Data from the HEROS trial in which patients with HER2+ sarcomas were treated with appropriately directed CAR products was published first in 2015 [22]. In this study, there were 17 evaluable patients with refractory, recurrent, or metastatic sarcomas (osteosarcoma, Ewing sarcoma, primitive neuroectodermal tumor, desmoplastic round cell tumor) treated with HER2-directed CAR T-cells without any pre-infusion lymphodepleting chemotherapy. For a median follow-up of 10.1 months (range, 1.1–37), they reported a median overall survival (OS) of 10.3 months (range, 5.1–29.1) with 4 patients achieving SD for greater than 12 weeks. Later, Hegde et al. reported on 13 patients with progressive, metastatic, or recurrent HER2+ sarcoma (osteosarcoma, rhabdomyosarcoma, synovial sarcoma) who were treated with HER2-directed CAR T-cells with varying lymphodepleting regimens [23]. By the end of the 6-week evaluation period, 21% of patients had a complete response (CR), 29% had SD, and 50% had progressive disease (PD). One patient with rhabdomyosarcoma even had a CR lasting >6 years at the time of publication and another patient with osteosarcoma had a CR lasting 42 months after which they relapsed. The median OS was 8.2 months (95% CI, 4.4–17) and median PFS was 2.4 months (95% CI, 1.3–8.7) at median follow-up of 8.2 months (range, 1.3–70.6).

Development of more novel CAR products is ongoing, predominantly in animal models. Cao et al. reported the development of a canine B7-H3-targeting CAR T-cell which dual expressed C-X-C motif chemokine receptor 2 (CXCR2), to improve migration to tumor sites [24]. Their study is of particular interest because high levels of B7-H3 expression has been observed in osteosarcoma and portends a poor prognosis [25]. Their study was done in a canine model with osteosarcoma and showed improved anti-tumor activity with their CAR product. Ongoing trials in humans with B7-H3-directed CAR T-cells are listed in Table 2. Another recent study by Xiao et al. investigated the use of FGFR4-directed CAR T-cells with an incorporated inducible caspase-9 (iCasp9) suicide gene system to enhance safety. CAR-T can be limited by severe adverse effects including cytokine release syndrome (CRS), immune effector cell associated neurotoxicity syndrome (ICANS), and on-target/off-tumor toxicity (OTOT) [26,27]. One proposed strategy to mitigate toxicity has been using the iCasp9 suicide gene system which allows for apoptosis of transduced CAR cells using the “trigger” drug AP1903 [28]. Their study showed that FGFR4-directed CAR T-cells had effective in vivo killing of rhabdomyosarcoma cells along with effective apoptotic activity using iCasp9 [27].

Although data on CAR T-cell therapy for sarcomas remains limited, early findings are encouraging. The generalizability of these trials is restricted due to a lack of diversity in patient age and functional status, as well as number of patients recruited. Encouragingly, early data suggest that CAR T-cells have an acceptable safety profile, which sets the stage for future research focusing on optimizing dosage and enhancing anti-tumor activity. Strategies for improving efficacy will include combining CAR T-cell therapy with ICIs, stimulatory cytokines, and other adjunct treatments. Another concern is mitigating off-tumor effects, where CAR T-cells recognize their target antigen on healthy tissues. Potential solutions include the use of inducible suicide genes to improve safety without compromising cytolytic activity. Overall, these studies have established CAR T-cells as a promising therapeutic option. As the field evolves, ongoing and future clinical trials will be paramount in improving safety, expanding targets, and improving efficacy (Table 2).

## 4. T-Cell Receptor (TCR) Gene-Modified T-Cell Therapy

Engineered TCRs are developed by obtaining T-lymphocytes via the leukapheresis of patient peripheral blood or tumor tissue. CD4^+^ and CD8^+^ T-cells are then transduced with a viral vector to express a T-cell receptor recognizing a specific peptide sequence presented by specific human leukocyte antigen (HLA) subtypes (Figure 3) [29]. TCR infusion is usually preceded by lymphodepleting chemotherapy. The potential benefit of engineered TCRs is that they allow for the recognition of intracellularly processed peptides presented via major histocompatibility complexes (MHCs). Sarcomas are typically immunologically cold, and natural-born T-cells do not survive in the TME to exert their effects against these tumors. Intracellular tumor antigens are derived from self-antigens, so immune cells that recognize these antigens are normally negatively selected against and do not proliferate [30,31,32]. Engineered TCRs are designed to have enhanced affinity for the target antigen, and the expansion of these populations in vitro allows for more effective targeting of otherwise poorly expressed peptide antigens. There are two heavily researched engineered TCRs to date: afami-cel and lete-cel.

One novel engineered TCR therapy for metastatic or unresectable SS and MRCLS is afamitresgene autoleucel (afami-cel). This is an HLA-restricted autologous CD4^+^ and CD8^+^ T-cell product transduced with a self-inactivating lentiviral vector to express an affinity-enhanced MAGE-A4-specific TCR. Melanoma-associated antigen A4 (MAGE-A4) is a cancer/testes antigen expressed in solid tumors including SS and MRCLS [33]. In terms of monitoring for safety considerations, there is concern that T-cell therapies can react to off-tumor sites such as similar epitopes on normal “healthy” cells. This is less of a concern for afami-cel because MAGE-A4 is expressed only in tumors and immune-privileged sites like the testes and placenta, but not in normal tissue [34].

Sanderson et al. published a preclinical evaluation of MAGE-A4-specific TCRs and found them to be highly potent without any major safety concerns [30]. In this study, afami-cel was infused into athymic mice with subcutaneous or xenografted tumors derived from human melanoma cell lines. They reported overall safety with very few off-target reactivities.

Hong et al. reported data on afami-cel in the clinical trial context, commenting on its efficacy and safety in a broader population of patients with solid tumors [35]. They reported on 38 patients with MAGE-A4^+^ relapsed or refractory metastatic solid tumors (including SS and MRCLS). Their patients underwent lymphodepletion with cyclophosphamide and fludarabine prior to infusion with T-cells. The overall response rate (ORR) was 24% (95% CI, 11.4–40.2) for all cancers and 44% (95% CI, 19.8–70.1) for SS, with an overall median PFS of 12.3 weeks (95% CI, 10.9–19.1) and 20.4 weeks (95% CI, 10.0–52.1) for SS. Median OS was 42.9 weeks (95% CI, 20.7–not reached) overall and 58.1 weeks (95% CI, 36.2–not reached) for SS. Three SS patients remained progression-free for >12 months. The median duration of response (DoR) was 25.6 weeks (95% CI, 12.28–not reached) overall and 28.1 weeks (95% CI, 12.28–not reached) for SS. That study showed early success, proving afami-cel has the potential to induce relatively long-lasting responses in solid tumors.

In the pivotal SPEARHEAD trial, 52 patients with metastatic or unresectable SS or MRCLS who had previously received treatment with an anthracycline or ifosfamide-containing regimen were given afami-cel following a regimen of lymphodepleting fludarabine and cyclophosphamide [29]. Over a median follow-up of 32.6 months, the ORR was 37% (95% CI, 24–51). The median DoR for patients with SS was 11.6 months (95% CI, 4.4–18.0). For MRCLS, it was 4.2 months (95% CI, 2.9–5.5). The response rates were higher in patients with higher MAGE-A4 expression and in patients with lower pre-treatment disease burden. The median PFS was 3.7 months overall (95% CI, 2.8–5.6) and 3.8 months (95% CI, 2.8–6.4) for patients with SS. The median OS was 15.4 months (95% CI, 10.9–28.7), with a survival probability of 60% at 12 months. For SS, the overall survival probability at 12 months was 90% and at 24 months was 70%. That study showed durable response in patients with SS as well as showed the efficacy of engineered TCRs in the treatment of these malignancies. In August 2024, afami-cel received accelerated approval from the FDA for the treatment of adults with unresectable or metastatic SS who had received prior chemotherapy.

Another anticipated therapy for SS and MRCLS is letetresgene autoleucel (lete-cel). This is also an autologous CD4^+^ and CD8^+^ T-cell transduced with an NY-ESO-1-specific TCR via a lentiviral vector. New York esophageal squamous cell carcinoma-1 (NY-ESO-1) is a testes cancer antigen which is expressed in a wide range of tumors, including 88% of MRCLS, 49% of SS, and 35% of myxofibrosarcomas [36]. Since NY-ESO-1 is expressed physiologically only in the testes and the testes do not display MHC class I molecules, these products can target tumors with minimal risk of off-target activity [37].

Robbins et al. published their findings from the first clinical trial for NY-ESO-1-directed engineered TCRs [37,38]. They reported on 11 patients with progressive metastatic melanoma and 6 patients with SS. The best responses included partial response (PR) in 4/6 SS patients, with the longest lasting 18 month. Further data from the same phase II study was reported 4 years later [39]. They reported on 18 patients at the time of publication with an ORR of 61% at 1 month follow-up with 10 PRs and 1 CR. They reported a few cases of long-lasting response, with one PR lasting greater than 20 months and another CR lasting 47 months at the time of publication. The estimated overall 3-year survival for sarcoma was found to be 38%, and 5-year survival was 14%.

D’Angelo et al. later reported on a pilot study of NY-ESO-1 engineered TCRs (termed SPEAR T-cells) for the treatment of histologically confirmed unresectable, or metastatic, progressive, persistent, or recurrent SS previously treated with ifosfamide or anthracycline-containing systemic chemotherapy [40]. They reported an ORR of 50% with 1 CR and 5 PRs, along with a median PFS of 15 weeks (range, 8–38) and a median OS of 120 weeks (range, 37–not reached) from 12 patients in one of the four cohorts studied. Further data from 42 patients in this trial were reported a year later [41]. Each cohort in this study received differing courses of pre-infusion lymphodepleting chemotherapy. In cohort 2, the responses included SD in 5 patients, PD in 1, and PR in 4. The median DoR was 10 weeks (range, 7.9–12.9). In cohort 4, the best responses included SD in 10 patients, PD in 1, and PR in 4. The median DoR was 16.3 weeks (range, 14.1–54.0). Cohort 3 was closed due to futility. This study was designed to evaluate the mechanisms of resistance and response. They found in some cohorts that responders had a significantly higher NY-ESO-1 vector copy number. They also noted that patients with higher doses of cyclophosphamide and fludarabine in their lymphodepleting regimen had better engraftment and persistence of SPEAR T-cells. There was some concern that antigen loss would be a mechanism of resistance against lete-cel, but an analysis of immunohistochemistry in the biopsies 8 weeks after infusion and >8 weeks after infusion found no change in the detection of NY-ESO-1 expression. The investigators also found evidence of long-lasting survival and functionality of these modified T-cells, with one tumor displaying evidence of persistent intra-tumoral lete-cel at 28 months post-infusion and another patient with lete-cel in peripheral blood with retained cytolytic activity at 12 months post-infusion.

Following the publication of that trial, Gyurdieva et al. published a post hoc analysis focusing on biomarkers that would be predictive for response and resistance to lete-cel [42]. There is already evidence that macrophages in the TME proport a poor prognosis, but they were investigating novel markers that could be used to predict response to lete-cel before, during, or after infusion [43]. Their study was based on 45 patients with unresectable, metastatic, or recurrent SS who received lete-cel infusion following a course of lymphodepleting chemotherapy. The clinical outcomes of patients in that study were detailed earlier in this section [40,41]. In that study, they found that responders across all cohorts had higher IL-15 levels pre-infusion compared to non-responders (*p* = 0.011). Responders also received a higher number of transduced memory CD8+ T-cells per kilogram (*p* = 0.039). Post-transfusion, the responders also had increased IFN-gamma, IL-6, and peak cell expansion compared to non-responders (*p* < 0.01, *p* < 0.01, *p* = 0.016, respectively). In an analysis of post-treatment tumor samples, they detected lete-cel infiltration and decreased expression of macrophage-related genes, which suggests some altering or remodeling of the TME. In addition, compared to prior to transfusion, biopsies of the tumors showed less expression of CD163 and CD68, which are cell surface markers associated with macrophages. A continued assessment of the cytokines, TME cellular populations, and expression of cell markers of interest will be needed to confirm these findings and establish reliable markers predictive of TCR therapy success.

IGNYTE-ESO is an ongoing phase two trial (NCT03967223) using lete-cel for the treatment of patients with SS or MRCLS who have failed previous anthracycline-based chemotherapy [44]. As of March 2023, 98 patients have been apheresed, 73 have received lete-cel, and 45 have been evaluated for efficacy. The reported ORR is 40% (95% CI, 20.3–62.3) with 2 CRs and 16 PRs. The median DoR was 10.6 months (95% CI, 10.6–not reached). Since the publication of these data, an update was presented at the Connective Tissue Oncology Society 2024 Annual Meeting, on 16 November 2024. Investigators reported 6 CRs and 21 PRs, along with a DoR of 12.2 months (95% CI, 6.8–19.5) and a median PFS of 5.3 months (95% CI, 4.0–8.0) [45]. There were no new or significant toxicities to report.

The most recent data come from a pilot study assessing lete-cel for patients with advanced MRCLS [46]. In total, 23 patients were enrolled in 2 cohorts. Cohort 1 received reduced-dose lymphodepleting chemotherapy, while cohort 2 received a standard dose. In cohort 1, the best overall responses were PR in two patients and SD in eight patients. In cohort 2, the best overall response was PR in four patients, SD in one patient, and PD in one patient. The reported ORR was 20% (95% CI, 2.5–55.6) and 40% (95% CI, 12.2–73.8) in cohorts 1 and 2, respectively. The median DoR was 5.3 months (95%CI, 1.9–8.7) and 7.5 months (95% CI, 6.0–not reached), and the median PFS was 5.4 months (95% CI, 2.0–11.5) and 8.7 months (95% CI, 0.9–not reached) in cohorts 1 and 2, respectively. It is important to note that cohort 1 had more patients with stage IV disease and a higher number of prior chemotherapy lines, which limits the conclusions that can be drawn from these data. Regardless, that study demonstrates the continued successes of lete-cel in the treatment of MRCLS with manageable adverse events. Lete-cel has since received FDA breakthrough therapy designation for the treatment of advanced MRCLS.

Another clinical trial investigated a different NY-ESO-1-specific T-cell therapy, TAEST16001, in 10 patients with SS and 2 with liposarcoma [47]. The patients received pre-infusion lymphodepleting chemotherapy with fludarabine and cyclophosphamide and low-dose systemic IL-2 after infusion. The trial reported an ORR of 41.7% (95% CI, 15.2–72.3) with PR in five patients, two of which lasted over one year. The median PFS was 7.2 months (95% CI, 2.5–11.8) with nearly all patients progressing at the time of data cutoff. The study showed that this specific product was well tolerated and induced a relatively long-lasting response.

There is also work already being done to improve the effectiveness of engineered TCRs, by combining them with other investigational therapies. Ishihara et al. recently reported a phase I trial of NY-ESO-1-specific TCRs combined with a lymph node-targeting nanoparticulate peptide vaccine for advanced STS [48]. This study used a pullulan nanogel:long peptide antigen (LPA) vaccine, which has been proven in preclinical models to recruit TCR T-cells to lymph nodes and tumor tissue [49]. The rationale behind the study is to see if this investigational vaccine can help overcome the limitations of the optimized delivery of TCR T-cells to tumor sites. The pullulan nanogel vaccine itself does not have any pharmacological activity and has been proven to be without side effects, alleviating concerns with adding another agent [50]. That study had two parts, with a preclinical phase confirming effective tumor killing and increased intratumoral TCR T-cells in a mouse model and then a clinical trial of three patients with SS treated with this investigational combination [48]. They reported best responses of SD in two out of the three patients, with long-term TCR T-cell persistence in one patient [48]. In other solid tumors, additional strategies being explored to improve engineered TCRs include incorporation of a CD8α co-receptor to be co-expressed with the engineered TCR in SPEAR T-cells [51].

While much of the preclinical and trial data for TCR T-cells is encouraging, especially for the treatment of SS and MRCLS, there is a great deal of additional investigation required. While TCRs carry the benefit of being able to recognize intracellularly processed antigens, one major limitation is that they are limited in their antigen recognition due to their restriction to certain HLA alleles. In the SPEARHEAD-1 trial, 373 patients with SS and MRCLS were pre-screened for HLA type eligibility and required MAGE-A4 expression levels, and only 105 (28%) were found to meet these eligibility criteria [29]. Current data are also limited because most studies have been single-armed, non-randomized, and non-blinded. Future studies will need to be powered to assess the superiority between engineered TCRs, other adoptive cellular therapies, and traditional systemic therapies. Additionally, more studies are needed to determine the optimal dosing of infused cells and lymphodepleting chemotherapy for improved safety and efficacy. Despite these challenges, there are encouraging findings from the above-mentioned trials for afami-cel and lete-cel, including the detection of persistent post-infusion T-cells in peripheral blood along with maintained HLA-directed tumor cell lysis months after infusion [35]. The continued development of cellular products with long-lasting functions and in vivo persistence will be essential for inducing sustained remissions. The current clinical trials are listed in Table 3.

## 5. Immune Microenvironment in Sarcomas

Sarcomas exhibit a highly heterogeneous TME, which plays a central role in shaping the response to adoptive cell therapies. Many subtypes are immunologically “cold”, characterized by low levels of T-cell infiltration, reduced MHC expression, and suppressive cytokines. Immune-suppressive cells like tumor-associated macrophages (TAMs), myeloid-derived suppressor cells (MDSCs), and regulatory T-cells are frequently enriched in the sarcoma TME, contributing to therapeutic resistance [31,32]. A prime worry is that this cellular environment bolsters resistance to physiologic anti-tumor immune response as well as immunologic therapies [52].

Solid tumors have been shown to alter the TME in order to escape the immune system through mechanisms like recruitment of tumor-promoting immune cells, alteration of antigens, and enhancement of immune-suppressive regulatory pathways [53]. Some of the immune cells which can promote tumor growth and persistence include regulatory T-cells, MDSCs, and TAMs. One of the challenges with sarcoma is the heterogeneity of the TME and infiltration with a varied population of both the aforementioned cells and immune-bolstering cells. This may explain the varied responses to ACT, as some tumors may be susceptible to ACT infiltration and cytotoxicity, while others may be more resistant. Some have suggested using ICIs as a way to mitigate the inherent resistance of the TME. However, in studies using ICIs for the treatment of various histological sarcoma subtypes, the clinical benefit is variable, which may be another manifestation of the heterogeneity of the immunological profile of the TME and the genetic profile of the individual tumor [53,54]. Future strategies to address the heterogeneity of sarcoma and the TME are discussed later.

There has been particular interest in TAMs. TAMs originate from MDSCs and exert pro-tumor effects via the release of angiogenic factors, thereby promoting oxygen and nutrient delivery while also secreting immunosuppressive cytokines [55,56]. These cells have two phenotypes. M1 is associated with an anti-tumor effect, while M2 is associated with a pro-tumor effect [57]. High levels of infiltration with M2 TAMs has been associated with worse prognosis in both STSs and osseous sarcomas [58,59].

B-cells are another population of interest in the TME. The role of B cells in inducing tumor immunogenicity is dependent on the associated maturity of tertiary lymphoid structures (TLSs). TLSs are ectopic lymphoid structures which form under states of persistent inflammation, like malignancy. Immature TLSs are characterized by B-cell populations which evolve as regulatory T-cells and have an immune-suppressive effect, whereas mature TLS B-cells differentiate into plasma cells and secrete anti-tumor antibodies [60]. An increased B-cell population in the TME has been shown to portend a favorable prognosis in sarcomas [61]. An analysis of STS tumors treated with pembrolizumab found that tumors that were deemed “immune high” due to a high expression of B-cell-related genes had a more favorable survival profile and had more response to immunotherapy [62].

Building on our understanding of the interactions within the TME, recent efforts have focused on therapies that modulate and enhance anti-tumor responses. Prospective targets include CSF1R, CD73, and CD47, which all promote anti-tumor effects. In fact, therapies using these targets have already been investigated with optimistic efficacy and safety profiles [63,64,65]. Unfortunately, some TME-targeting therapies have been shown to have significant drug toxicity, limiting their potential applicability [66]. An active clinical trial is investigating the use of a CSF1R inhibitor administered alongside immunotherapy for patients with advance high-grade sarcomas (NCT04242238).

## 6. Challenges and Limitations

### 6.1. Safety Concerns and Adverse Events

As with most newly developed therapeutics, safety is of paramount concern. The inherent side effects of ACTs alongside those of adjunct lymphodepleting chemotherapy or other systemic therapies can be significant and must be intensively characterized to assess the pre-treatment risk for each patient. While acute or subacute side effects are of primary concern, some ACT products have been shown to persist in peripheral blood and/or tumors for months to years, raising concerns for long-term exposure and cumulative toxicity.

As already discussed, clinical trial data on TILs for sarcoma are fairly limited; however, the most common side effects described were cytopenias related to lymphodepleting chemotherapy, along with fevers related to TIL infusion [14]. Cytopenias were also common for engineered TCR T-cells and CAR T-cells [23,35,37,40,44]. The vast majority of these were related to lymphodepleting chemotherapy and were transient, without complication.

Cytokine release syndrome (CRS) is another significant consideration. In the IGNYTE-ESO trial, they reported that 89% of patients had CRS [44]. The SPEARHEAD trial reported CRS in 71% of patients, with only one case of grade 3 CRS [29]. In the HEROS trial, 79% of patients had CRS, with one reported case of grade 4 complicated by respiratory failure, progressive pulmonary metastatic disease, and eventual death [23]. In the same study, they had another case of grade 3 CRS, with respiratory involvement in the same treatment cohort leading to premature termination of that specific cohort in which patients received a higher dose of CAR^+^ T-cells at initial infusion [23].

Along with CRS, another feared toxicity of cellular therapy is ICANS and other neurological sequelae. Fortunately, there has yet to be any reports of significant incidences of ICANS in ACTs for sarcoma [47].

Hong et al. reported less common but significant adverse events in patients receiving MAGE-A4-directed TCRs [35]. They described two possible treatment-related deaths, both occurring in the cohort receiving the most intense lymphodepleting regimen. One patient died of aplastic anemia, and another due to stroke [35]. While these specific adverse events did not occur in other studies of MAGE-A4-directed TCRs, these on-study deaths are significant, and patients should continue to have their cell counts and neurological exams monitored carefully following infusion.

Another broad concern is what has been deemed as “off tumor, on target” toxicity. An ideal cellular product would target antigens or epitopes only present on target (i.e., malignant) cells, and not on healthy cells. Particularly in solid tumors, many cell surface or intracellular antigens are expressed in both tumors and healthy tissues, leading to potential “off-tumor” toxicities. This manifestation has been described in prior solid tumors leading to fatal toxicities related to respiratory failure [67,68]. Notably, for the aforementioned engineered TCR-based therapies, adverse effects are limited due to the lack of NY-ESO-1 and MAGE-A4 expression in normal healthy tissues [37]. A continued investigation of similar expression-limited targets will be essential to expand the treatment options to other sarcoma subtypes and solid tumors. In contrast, for CAR-T cells, there remains a significant risk of off-tumor toxicity due to the target antigens/epitopes being expressed more commonly in healthy tissue. In the HEROS trial investigating HER-2 directed CAR-T, they described one death related to respiratory failure but deemed it to be due to pulmonary progression of the primary malignancy rather than a direct effect of the CAR T-cells [23]. One promising strategy of mitigating the risk of off-tumor toxicity while still producing ACTs for high-yield targets is employing suicide genes or motifs like iCasp9, as used by Xiao et al., to decrease the activity of ACTs when off-tumor activity is suspected [27].

Some have suggested mitigating adverse events by using lymphodepleting chemotherapy-sparing regimens similar to Ishihara et al. as described earlier [48]. This is done with the hope of decreasing the risks of CRS and secondary malignancies. Another proposed strategy is to use treatment regimens without IL-2, such as D’Angelo et al., who posited that their IL-2-free regimens led to decreased toxicities compared to prior trials [40].

### 6.2. Predictive Markers of Success

Another chief need in the development of ACTs is reliable markers of persistence of cells and reliable predictors of response. Some recent studies commented on specific metrics that they found to be associated with response. Pan et al. reported that the expansion of NY-ESO-1-specific T-cells as detected by qPCR was higher in responders with five patients with detectable qPCR up to 25 weeks post-infusion, with the longest expression time being 9 months [47]. D’Angelo et al. also found that responders had significantly higher vector copy numbers of NY-ESO-1 compared to non-responders (*p* = 0.0411) [40]. Gyurdieva et al. reported that patients with lymphodepleting regimens containing fludarabine had higher IL-15 levels, which was associated with response to lete-cel [42]. These metrics of copy number, qPCR assays, and IL-15 levels could represent reasonable serum markers to follow in patients to predict response to therapy at early stages post-infusion.

It has been suggested that ACT may eventually become ineffective in patients due to the down-regulation of genes used for antigen presentation, effectively removing the target of these medications [69]. Ramachandran et al. explored this idea using a gene panel associated with antigen presentation and found no significant alterations in expression as a sign of development of resistance to lete-cel [41]. Another concern is the decreased expression of cellular targets, which was described recently by D’Angelo et al., reporting a few patients with loss or decreased tumor NY-ESO-1 expression following treatment with lete-cel for MRCLS [46]. Loss of expression has not been found to be a major concern in previously reported data on lete-cel or other cellular therapies for sarcoma. These findings should be followed in further studies to determine if they represent a significant mechanism of resistance. Exhaustion markers are also another metric to follow as activated T-cells can have declining function secondary to chronic stimulation [70]. Exhausted T-cells are typically found to have reduced cytokine production along with the expression of inhibitory receptors such as LAG-3, PD-1, and CTLA4 [70]. D’Angelo et al. encouragingly found that lete-cel remained negative for PD-1 and LAG3 during the post-infusion analysis period [40]. The prevention of T-cell exhaustion will be crucial for maintaining longer-lasting cellular therapies, reducing the need for repeated infusions and exposures.

### 6.3. Limitations

Despite promising advances, ACT faces several limitations to its broader efficacy. The TME hampers therapeutic outcomes through immunosuppression, antigen escape, and cell exhaustion. Moreover, distinct challenges specific to TILs, CAR T-cells, and TCR therapies—including difficulties in cell expansion, limited infiltration, safety concerns, and variability in antigen expression—further complicate treatment efficacy.

As discussed in the prior section, the TME is a major limiting factor for ACT. The TME dampens immune responses via decreased chemokine expression and increased population of inhibitory T-cells, which inherently limit the effectiveness of ACT [71].

The need to collect TILs directly from peripheral blood or directly from the tumor makes them a very individualized product, leading to inherent challenges in maintaining a standard in quality, unlike CAR-T products, which can be produced on a much larger scale. TILs also require a large number of lymphocytes to be infused, which requires extended in vitro growth, which may be undesirable if the disease in question is rapidly progressing. Another concern is TIL exhaustion, which can occur due to prolonged expansion times in vitro and has been proven to lead to poor cytotoxicity and persistence of TILs [72]. Another challenge is that apheresed TILs are a clonally heterogenous population that may recognize an equally heterogenous tumor. This means that expanded and transfused cells may be effective against a few tumor-associated antigens but could potentially leave a population of tumor cells unharmed, leading to incomplete responses.

Due to the large body of research conducted on CAR T-cells, there is more information on their limitations. Some major topics when considering CAR products would be their unimpressive outcomes in solid tumor malignancies, poor localization and entry to tumor tissues, resistance of the TME, safety concerns (esp. CRS and ICANS), difficulties in maintaining cytotoxic activity and persistence, challenges in preventing off-target toxicity, challenges in producing universal “off the shelf” CAR T-cells for solid tumors, and target selection as many solid tumor antigens are expressed in otherwise normal tissue as well. Specifically for CAR-T, there are concerns about safety, especially as it pertains to CRS and ICANS. These entities have been observed in trials of CAR-T therapy for sarcoma, as discussed earlier in this review.

A few additional limitations apply to all the cellular products listed in this review. One is the issue of infiltration. Sarcomas, like other solid tumors, have particularly dense extracellular matrices, making it difficult for these cells to reach their targets [73,74,75]. Another consideration is cell exhaustion. Prolonged in vitro expansion, persistent stimulation of therapeutic cells both in vitro and in vivo, and the effects of the inhibitory TME can all result in exhaustion of ACT and loss of function [76,77,78]. Additionally, TCR T-cells, during in vitro expansion, typically are terminally differentiated, leading to loss of the adaptive capabilities of other T-cell subsets, hampering their ability to induce a long-standing response [72,79]. Attempts have been made to overcome this limitation by selectively eliminating immune-suppressive regulatory T-cells and by using T-cell products containing a larger subset of less differentiated cells prior to in vitro expansion [80].

Another concern is antigen escape. ACT relies on the recognition of tumor markers, so their effect is limited on tumors with low antigen density. Due to the inherent heterogeneity of solid tumor antigen expression, there can be significant variability of target antigen expression within a given tumor. This leads to the phenomenon of antigen escape, where tumor cells with low target antigen expression survive initial treatment and persist, making full eradication of a tumor by ACT challenging [81].

The limitations of the reported data in these trials are equally important to address. Most studies are early-phase clinical trial or preclinical models with small sample sizes. In addition, due to the targeted nature of these therapies, trial sizes are limited due to the limited expression of tumor antigens by the sarcoma in question. Many studies lack standardized endpoints, long-term follow-up, and comparison arms, which further complicate the applicability of these therapies to real-world scenarios and limit statements on safety and efficacy. These limitations pose significant challenges in generating generalizable conclusions across diverse sarcoma subtypes and patient populations.

### 6.4. Scalability and Manufacturing

ACTs demand a personalized and labor-intensive development process for each product, requiring considerable time and cost investments. As research advances, we may see more facilities equipped to produce ACTs with more efficient processes. However, owing to their complexity and novelty, these treatments will need to be highly regulated and only administered at experienced institutions with strict protocols. These restrictions will likely limit their availability to academic centers for quite some time.

## 7. Future Directions

The field of ACT for sarcoma is rapidly evolving. While promising results have been seen with engineered TCRs and CAR T-cells in subtypes such as synovial sarcoma and MRCLS, durable responses remain limited to a subset of patients. A key focus of future work is the identification of novel, tumor-restricted antigens with high expression across a broader array of sarcoma subtypes. Combination approaches are also gaining traction, particularly those pairing ACT with immune checkpoint blockades, oncolytic viruses, or stimulatory cytokines such as IL-15. For example, trials are currently exploring the synergy of PD-1 inhibitors with CAR T-cell or TIL therapy [53]. Finally, resistance mechanisms, including antigen loss, clonal heterogeneity, and T-cell exhaustion, are being addressed with strategies like multiplexed targeting, logic-gated CARs, and engineered T-cells with resistance to inhibitory signals. Overcoming these barriers will be essential for ACT to achieve broader, more durable impacts across the diverse sarcoma landscape.

One area of interest is the potential for combination therapies. There is a lack of data for these approaches in sarcoma, but they have been studied in other tumors, with preclinical models showing some success. One study looked at the combination of TCR T-cells with PD-1 monoclonal antibodies in a mouse model for lung cancer and found improved efficacy of ACT with this combination [82]. Another study found that the addition of PD-1 monoclonal antibodies to anti-CEA CAR T-cells in liver metastases in a mouse model led to less CAR exhaustion [83].

Chemokines secreted by tumors play a role in T-cell migration and infiltration into solid tumors. One strategy to enhance T-cell homing has included the transduction of chemokines and immune-stimulating cytokines in CAR-T cells [84,85,86]. Immunostimulatory cytokines impact the function and persistence of T-cells, though toxicity is a limitation of systemic administration. The expression of IL-7 by CAR T-cells enhanced activity in a liver cancer model, and the co-expression of IL-12 by modified TCR T-cells may be associated with improved anti-tumor immune response [87,88].

While significant developments for TILs and TCR T-cells have been limited, there is a fair amount of prospective work being done on CAR products. An area of interest is logic-gated CARs, which act somewhat like circuits, allowing the use of multiple activating and inhibitory signals to allow for accurate recognition of tumors that would otherwise be difficult to target given their antigen heterogeneity [89]. One example of these products is an “OR” logic-gated CAR, which involves the use of receptors with multiple antigen-recognition domains, allowing for the activation of T-cells by multiple tumor-associated antigens. This type of product has already shown some efficacy in B-cell malignancies [90,91]. Another type is an “AND” logic-gated CAR, which requires two antigens to simultaneously be recognized for the T-cell to be activated, thereby increasing the specificity of the cellular product and decreasing off-tumor toxicity. One other type is the “AND-NOT” logic-gated CAR, which allows for recognition of a target antigen along with another antigen-recognizing motif tied to an inhibitory signal. The theory behind this product is that it would allow for on-target effects while limiting off-tumor effects if antigens on normal cells activate the inhibitory signal. Another possible future direction involves the use of oncolytic viruses to infect tumor cells and increase the expression of potential target tumor antigens along with CAR products that recognize these antigens. This has already been investigated in mice models with CD19-targeting CARs, which showed effective targeting of solid tumors using oncolytic viruses [92]. This strategy could help overcome antigen escape, by inducing the expression of target antigens in tumor cells that otherwise would have low expression and evade ACT.

Another strategy to directly overcome the limitations of the immune-suppressive TME is bispecific T-cell engagers. This product uses CARs with one domain targeting tumor surface antigens and another mediating attachment to T-cells to potentiate anti-tumor effects. To improve infiltration through a dense extracellular matrix of solid tumors, there has been research into engineered CARs which secret heparinase, which has been proven to more effectively penetrate the extracellular matrix in preclinical models, but there were reported intolerable toxicities in clinical trials [75,93].

One additional strategy is using the inherent conditions of the TME like hypoxia and acidic conditions to selectively activate CAR T-cells, reducing the chance of activation under physiologic conditions in “normal” tissues. One example of this was a CAR product tied to an oxygen degradation domain, leading to the expression of the CAR in hypoxic conditions and degradation with normal oxygen levels. This product was studied in a murine model, showing effective tumor suppression without significant systemic toxicity [94].

A major concern is the discovery of novel target antigens for ACT. An antigen landscape analysis involves characterizing tumor-associated antigens to identify actionable targets. By integrating genomic, transcriptomic, and proteomic data, this approach facilitates the discovery of neoantigens, which can enhance the specificity and efficacy of engineered T-cell products. Comprehensive profiling enables more personalized ACT strategies, potentially overcoming the heterogeneous antigen expression that limits treatment response in sarcomas [95,96,97]. Furthermore, understanding antigen presentation patterns through a landscape analysis is crucial to predicting immune escape mechanisms and developing therapies to improve long-term therapeutic outcomes.

## 8. Conclusions

As the field of oncology develops alternatives to cytotoxic systemic chemotherapy, ACT represents an exciting and novel field with potential to induce meaningful improvements in the survival and quality of life in patients with previously untreatable, resistant sarcoma or other solid tumors. These modalities have already shown encouraging success in both preclinical models and clinical trials alike, with acceptable adverse events. Although these products have been further developed, critical topics remain, such as optimal dosing, frequency of repeat infusions, mechanisms of resistance, predictors of treatment success, long-term benefit and adverse events, and potential for combination with other targeted therapy or traditional systemic chemotherapy. Future laboratory and clinical literature will reveal whether these agents can safely and reliably induce long-lasting responses with improved quality of life in patients with these otherwise aggressive and lifespan-limiting malignancies.

## Figures and Tables

**Figure 1 cancers-17-01302-f001:**
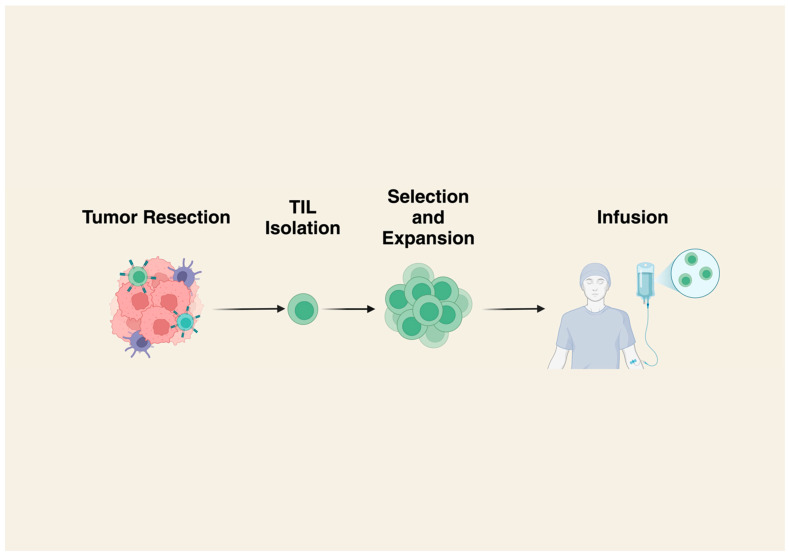
Production of TILs. Tumor tissue is first resected or biopsied, and tissue is then processed to isolate TILs. Lymphocytes showing strong anti-tumor activity are selected and expanded in vitro. Finally, patients receive lymphodepleting chemotherapy followed by TIL infusion. Created in BioRender.com. https://BioRender.com/k76w677.

**Figure 2 cancers-17-01302-f002:**
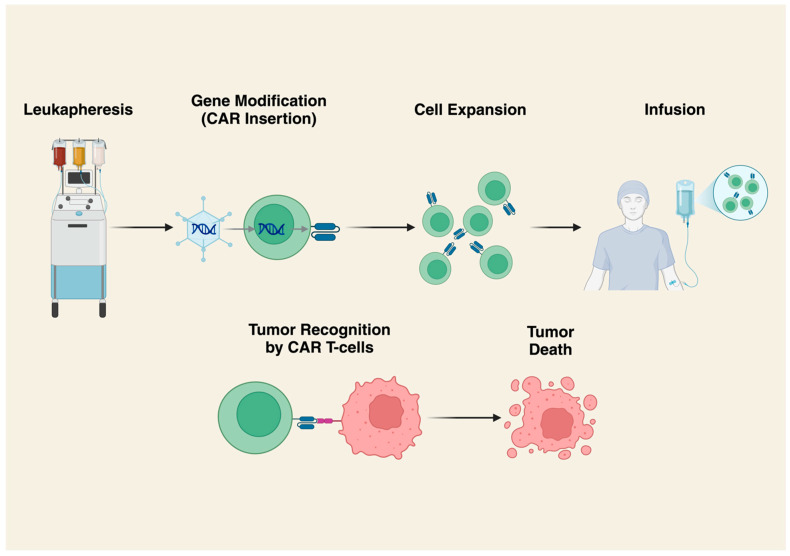
Production and Mechanism of Action of CAR T-Cells. T lymphocytes are first collected by leukapheresis, then a CAR construct is added via a viral vector. Cells are then expanded in vitro. Following cell expansion and quality control checks, patients receive lymphodepleting chemotherapy followed by CAR T-cell infusions. Cytotoxic activity is achieved by recognition of cell surface antigens by CAR T-cells leading to immune-mediated destruction of tumor cells. Created in BioRender.com. https://BioRender.com/q22d481.

**Figure 3 cancers-17-01302-f003:**
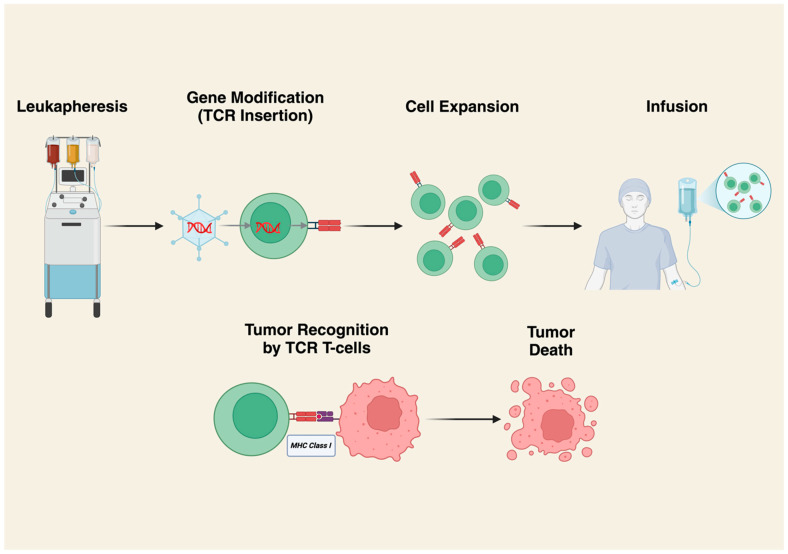
Production and mechanism of action of TCR gene-modified T-cells. T-cells are first isolated by leukapheresis; then, a tumor-specific TCR gene is introduced using a viral vector. The cells are then expanded in vitro and rigorously assessed for safety and efficacy with quality control measures. Following lymphodepletion, patients receive infusions of TCR-modified T-cells. Cytotoxic activity is achieved by recognition of MHC class I molecules displaying the target intracellular peptide leading to immune-mediated destruction of tumor cells. Created in BioRender.com. https://BioRender.com/e59a255.

**Table 1 cancers-17-01302-t001:** Active or completed adoptive TIL trials for sarcomas.

Trial	Cancer Treated *	Adoptive Cellular Product	Status **
NCT04052334	Soft-Tissue Sarcoma	TIL	Completed
NCT05607095	Undifferentiated Pleomorphic Sarcoma Dedifferentiated Liposarcoma	TIL	Recruiting
NCT03725605	Soft-Tissue Sarcoma	TIL	Completed
NCT03935893	Solid Tumor	TIL	Recruiting
NCT06566092	Rhabdomyosarcoma Ewing Sarcoma	TIL	Recruiting
NCT03449108	Bone Sarcoma Soft-Tissue Sarcoma	TIL	Active
NCT03449108	Bone Sarcoma Soft-Tissue Sarcoma	TIL	Active

To the best of our knowledge, this table contains all active, completed, and recruiting clinical trials using TILs for the treatment of sarcomas accessible on ClinicalTrials.gov. Sarcomas treated in the trials are listed if specified by subtype in the study’s online entry; however, many trials recruited participants with other malignancies (may be found via respective trial numbers).

,

* When abundantly clear, we include in this column the sarcoma subtype that was assessed in the clinical trial. However, many trials investigated solid tumors broadly, including sarcomas. These studies are marked as such in this column. ** This table may not be exhaustive as there are trials listed with statuses of “unknown” and “not yet recruiting” that were excluded.

**Table 2 cancers-17-01302-t002:** Active or completed adoptive CAR T-cell trials for sarcoma.

Trial	Cancer Treated *	CAR Target	Status **
NCT04995003	Sarcoma	HER2	Recruiting
NCT00902044	Sarcoma	HER2	Active
NCT04511871	Solid Tumor	HER2	Active
NCT03635632	Solid Tumor	GD2	Active
NCT03373097	Solid Tumor	GD2	Recruiting
NCT03721068	Osteosarcoma	GD2	Recruiting
NCT04539366	Osteosarcoma	GD2	Recruiting
NCT04483778	Solid Tumor	B7-H3	Active
NCT06500819	Solid Tumor	B7-H3	Recruiting
NCT04556669	Solid Tumor	CD22	Recruiting
NCT06087341	Sarcoma	NKG2D	Recruiting
NCT05312411	Osteosarcoma	FITC-E2	Active
NCT05103631	Solid Tumor	GPC3	Recruiting
NCT04377932	Solid Tumor	GPC3	Recruiting
NCT04715191	Solid Tumor	GPC3	Recruiting
NCT05120271	Solid Tumor	GPC3	Recruiting

To the best of our knowledge, this table contains all active, completed, and recruiting clinical trials using CAR T-cells for treatment of sarcomas accessible on ClinicalTrials.gov. Sarcomas treated in the trials are listed if specified by subtype in the study’s online entry, however, many trials recruited participants with other malignancies (may be found via respective trial numbers).

,

* When abundantly clear, we include in this column the sarcoma subtype that was assessed in the clinical trial. However, many trials investigated solid tumors broadly, including sarcomas. These studies are marked as such in this column. ** This table may not be exhaustive as there are trials listed with statuses of “unknown” and “not yet recruiting” that were excluded.

**Table 3 cancers-17-01302-t003:** Active or completed TCR gene-modified T-cell trials for sarcoma.

Trial	Cancer Treated *	TCR Target	Status **
NCT04044768	Synovial Sarcoma Myxoid/Round Cell Liposarcoma	MAGE-A4	Recruiting
NCT03132922	Synovial Sarcoma Myxoid Round Cell Liposarcoma	MAGE-A4	Active
NCT05642455	Synovial Sarcoma Osteosarcoma	MAGE-A4	Recruiting
NCT06703346 (Sub-Study of NCT03967223)	Synovial Sarcoma Myxoid Round Cell Liposarcoma	NY-ESO-1	Active
NCT05993299 (Sub-Study of NCT03967223)	Synovial Sarcoma Myxoid Round Cell Liposarcoma	NY-ESO-1	Active
NCT01343043	Synovial Sarcoma	NY-ESO-1	Completed
NCT01477021	Synovial Sarcoma	NY-ESO-1	Completed
NCT06083883	Synovial Sarcoma Myxoid Round Cell Liposarcoma	NY-ESO-1	Recruiting
NCT02650986	Synovial Sarcoma	NY-ESO-1	Active
NCT03450122	Synovial Sarcoma Myxoid Round Cell Liposarcoma	NY-ESO-1	Completed
NCT03967223	Synovial Sarcoma Myxoid Round Cell Liposarcoma	NY-ESO-1	Active
NCT03250325	Synovial sarcoma	NY-ESO-1	Completed
NCT05296564	Synovial Sarcoma Soft-Tissue Sarcoma	NY-ESO-1	Recruiting
NCT02992743	Myxoid Round Cell Liposarcoma	NY-ESO-1	Completed
NCT04318964	Soft-Tissue Sarcoma	NY-ESO-1	Active
NCT03462316	Bone Sarcoma Soft-Tissue Sarcoma	NY-ESO-1	Active
NCT02319824	Sarcoma	NY-ESO-1	Completed
NCT03450122	Synovial Sarcoma Myxoid Round Cell Liposarcoma	NY-ESO-1	Completed
NCT02869217	Synovial Sarcoma	NY-ESO-1	Active
NCT05621668	Bone Sarcoma Soft-Tissue Sarcoma	Attil12	Recruiting

To the best of our knowledge, this table contains all active, completed, and recruiting clinical trials using TCR gene-modified T-cells for the treatment of sarcomas accessible on ClinicalTrials.gov. The sarcomas treated in the trials are listed if specified by subtype in the study’s online entry; however, many trials recruited participants with other malignancies (may be found via respective trial numbers).

,

* When abundantly clear, we include in this column the sarcoma subtype that was assessed in the clinical trial. However, many trials investigated solid tumors broadly, including sarcomas. These studies are marked as such in this column. ** This table may not be exhaustive as there are trials listed with statuses of “unknown” and “not yet recruiting” that were excluded.

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
