# Peer review of "Review of Adoptive Cellular Therapies for the Treatment of Sarcoma"

_cancers, 2025, doi:10.3390/cancers17081302_

Round 1
Reviewer 1 Report
Comments and Suggestions for Authors
The authors Fradin and Charlson provide an overwiew on the adoptive cellular therapies for the treatment of sarcoma.
The review is well organized and provide a useful summary of the recent advancement in the field of immunotherapy in the landscape of sarcoma.
The manuscript would benefit from the followings
- Latest Who should be included: WHO Classification of Tumours Editorial Board. WHO Classification of Tumours of Soft Tissue and Bone, 5th ed.” Lyon, France: IARC Press; 2020
- The followings studies should be referenced for proper discussion:
Clinical and translational implications of immunotherapy in sarcomas. Front Immunol. 2024 Jun 25;15:1378398. doi: 10.3389/fimmu.2024.1378398. PMID: 38983859; PMCID: PMC11231074.
Immunotherapy in Sarcoma: Current Data and Promising Strategies. Am Soc Clin Oncol Educ Book. 2024 Jun;44(3):e432234. doi: 10.1200/EDBK_432234. PMID: 38781557.
Current Landscape of Immunotherapy for Advanced Sarcoma. Cancers (Basel). 2023 Apr 13;15(8):2287. doi: 10.3390/cancers15082287. PMID: 37190214; PMCID: PMC10136499.
- Study limitations should be included
Reviewer 2 Report
Comments and Suggestions for Authors
This review is devoted to an urgent and promising topic - the application of adoptive cell therapies in the treatment of sarcomas. The authors of the review examine in detail three main approaches of adoptive cell therapies: tumour-infiltrating lymphocyte (TIL) therapy, CAR T-cell therapy, and therapy using T cells genetically modified to express specific T-cell receptors (TCRs). Each of these areas is analysed with respect to current advances, challenges, and future prospects. Sarcomas remain a significant challenge for oncologists, and the exploration of new therapies, including adoptive cell therapies, represents a crucial area of research. The review highlights the potential of these techniques, making it a valuable resource for the scientific and medical community. Nevertheless, there are a number of issues that preclude an entirely positive assessment:
-
The authors do not describe the features of the immune microenvironment in sarcomas, particularly the heterogeneity of the microenvironment, which plays a key role in the efficacy of adoptive cell therapies.
-
While the review covers various sarcoma types, the authors do not provide a detailed analysis for each type.
-
The discussion of combination approaches is limited. Although the authors briefly mention the possibility of combining ACT with other therapies, this area warrants more detailed exploration, given its potential to overcome tumour resistance.
-
The review lacks a critical analysis. A more critical assessment of the current limitations and risks of ACT would have provided readers with a clearer understanding of the realistic expectations for these techniques.
Reviewer 3 Report
Comments and Suggestions for Authors
This manuscript reports the adoptive cell therapy approaches for the treatment of sarcomas. The authors listed several clinical trials pointing out the pros and cons of these therapeutic approaches.
The manuscript is well written. The flow of the information is good and the topic is of interest.
The figures are simple and clear but really basic. The possible future developments are not specified and considered by the authors.
However, this manuscript gives a little on the topic itself. The main point is the difficulty of finding antigens or good targets for the treatment of sarcomas. This manuscript does not consider possible novel therapies based on ACT. On one hand, the manuscript is good but it is limited, although it is updated.
Without sections on antigens landscape analysis in sarcomas and novel approaches for adoptive T-cell therapies, this manuscript is too limited to be accepted for publication. I understand the clinical point of view of the authors, but this present form is better for another more clinical scientific journal.
Reviewer 4 Report
Comments and Suggestions for Authors
This paper is a valuable compilation of the current adoptive cellular therapy (ACT) landscape for sarcomas. Completed and ongoing trials are presented and commented on. Open questions related to ACTs are pointed out. The paper is suitable for physicians who are involved in sarcoma therapy and who want to get an overview of the current development of ACT products in this field.
minor comment:
- You could perhaps also point to resistance mechanisms in TIL-based therapy that may arise due to clonal heterogeneity
Round 2
Reviewer 3 Report
Comments and Suggestions for Authors
The authors replied to the reviewers' queries.